# X-MEN: Guaranteed XOR-Maximum Entropy Constrained Inverse Reinforcement Learning

**Fan Ding**[1]                    **Yexiang Xue**[1]

[1]Department of Computer Science, Purdue University, West Lafayette, Indiana, USA.

## Abstract

Inverse Reinforcement Learning (IRL) is a powerful way of learning from demonstrations. In this paper, we address IRL problems with the availability of prior knowledge that optimal policies will never violate certain constraints. Conventional approaches ignoring these constraints need many demonstrations to converge. We propose XOR-Maximum Entropy Constrained Inverse Reinforcement Learning (X-MEN), which is guaranteed to converge to the global optimal reward function in linear rate w.r.t. the number of learning iterations. X-MEN embeds XOR-sampling – a provable sampling approach which transforms the #-P complete sampling problem into queries to NP oracles – into the framework of maximum entropy IRL. X-MEN also guarantees the learned IRL agent will never generate trajectories that violate constraints. Empirical results in navigation demonstrate that X-MEN converges faster to the optimal rewards compared to baseline approaches and always generates trajectories that satisfy multi-state combinatorial constraints.

## 1 INTRODUCTION

Inverse Reinforcement Learning (IRL) [Ng and Russell, 2000, Abbeel and Ng, 2004, Ziebart et al., 2008, Arora and Doshi, 2021, Li, 2017] provides an important way to learn from demonstrations. IRL assumes that the demonstrator implicitly maximizes the cumulative reward of a Markov Decision Process (MDP). The goal of IRL is to recover the unknown reward function from the observed demonstrations. Various IRL algorithms have been proposed, including Linear IRL [Ng and Russell, 2000, Abbeel and Ng, 2004] and Large-Margin Q-Learning [Ratliff et al., 2006]. To differentiate among multiple reward functions which lead to similar behaviors, Maximum Entropy Inverse Reinforcement Learning (MaxEnt IRL) [Ziebart et al., 2008, Wulfmeier et al., 2015, Finn et al., 2016, Ho and Ermon, 2016] assumes that the demonstrator samples trajectories from a maximum entropy distribution parameterized by the cumulative reward.

In this paper, we focus on IRL problems where certain constraints are known beforehand and hence do not need to be rediscovered by the learning algorithm. The trajectories from the demonstrator are known to satisfy these constraints and we require the IRL agent to satisfy these constraints as well. Indeed, standard IRL algorithms [Abbeel et al., 2007, Vasquez et al., 2014, Scobee et al., 2018] can be applied to this scenario without modifications and they eventually discover the optimal reward function, which generates trajectories satisfying all constraints. Nevertheless, it may require a large amount of demonstrations to learn these constraints. Worse still, it is still possible for the IRL agent to produce trajectories which occasionally violate constraints even after many training epochs. This is especially problematic in safety critical domains, such as autonomous driving, robotic surgery, etc.

Recent work has attempted to embed constraints into IRL. For example, the work of [Vazquez-Chanlatte et al., 2017, Kalweit et al., 2020] uses demonstrations to learn a rich class of possible specifications that can represent a task. Others have focused specifically on learning constraints, that is, behaviors that are expressly forbidden or infeasible [Chou et al., 2018, Subramani et al., 2018, McPherson et al., 2018, Scobee and Sastry, 2019, Anwar et al., 2020, McPherson et al., 2021]. Nevertheless, so far the attempts have been focused on *single-state* constraints, where a handful of actions are forbidden in certain states and these forbidden actions have little impact for future state-action transitions. Their approaches cannot address *multi-state combinatorial* constraints, which limits a chain of actions spanning multiple time stamps. For example, Figure 1 (c) demonstrates a navigation task where constraints require at least half of the states in each trajectory is located in the shaded area. With this constraint imposed, only trajectories passing the

*Accepted for the 38th Conference on Uncertainty in Artificial Intelligence* (UAI 2022).

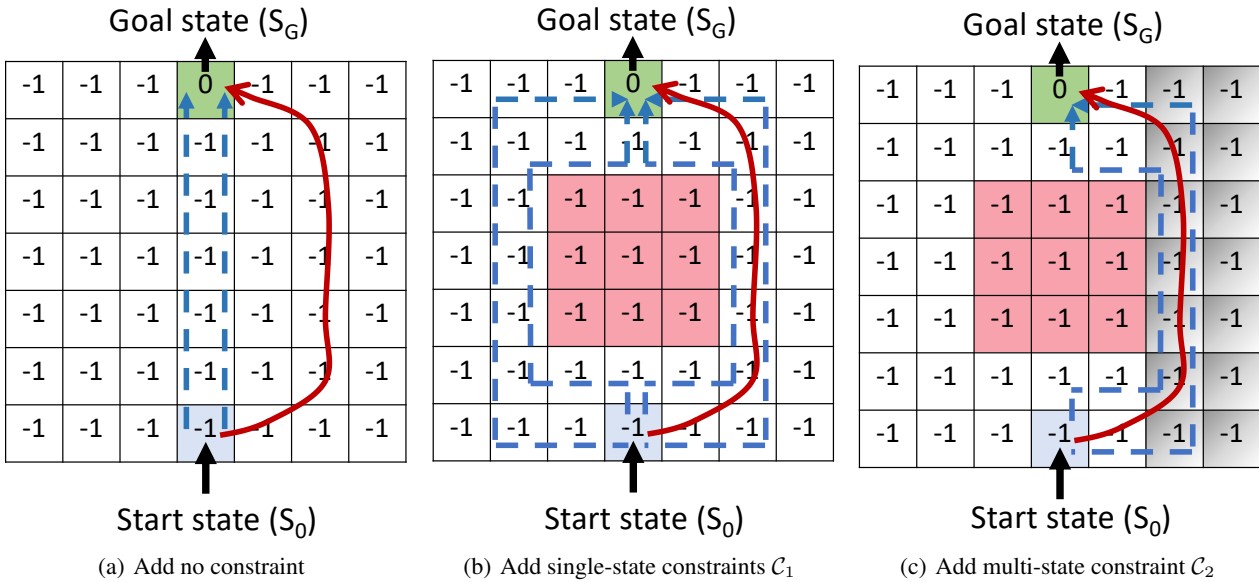

Figure 1: Examples of constrained IRL problems. The agent wants to move from the start state $S_0$ (blue grid) to the goal state $S_G$ (green grid). Ground truth demonstration is shown in the red line. The same initial reward function before learning is used for all 3 situations, with one-step reward listed in each grid. Most likely trajectories under the initial reward function (e.g.,those maximizing rewards and subject to constraints) are shown using blue dashed arrows. (a) When no constraint is added to the MDP, the agent finds the shortest path directly upward from $S_0$ to $S_G$. (b) When single-state constraint $\mathcal{C}_1$, which forbids the agent to to pass through the red grids, is imposed, the agent can detour from either the left or the right side. (c) When there are an additional multi-state constraint $\mathcal{C}_2$ imposed, which constrains at least half of all passing states in the shaded area, the optimal trajectory is to detour from the right side. Notice that this behavior aligns with the demonstration.

right-hand side are possible. Such constraints cannot be addressed with previous approaches, which mask out actions from certain states.

In this work, we propose **X**OR-**M**aximum **EN**tropy (X-MEN) Constrained Inverse Reinforcement Learning, which ***provably converges to the optimal reward function for MaxEnt IRL in linear number of training steps***, even in the presence of hard combinatorial constraints. X-MEN also guarantees to produce trajectories which satisfy multi-state combinatorial constraints. X-MEN is based on the Maximum Entropy IRL learning [Ziebart et al., 2008, Boularias et al., 2011]. Distinctively, X-MEN harnesses XOR-sampling to estimate the gradient of the expected reward from the current model distribution. The recently proposed XOR-Sampling [Gomes et al., 2007a, Ermon et al., 2013a,b] reduces the sampling problem into queries of NP oracles via hashing and projection, and guarantees a constant factor approximation for the expectation estimation. To maximize the likelihood of the demonstrated behavior, X-MEN uses Stochastic Gradient Descent (SGD) to maximize the difference between expected reward from the demonstration and that from the trajectories sampled from the current model distribution, a procedure closely resembling contrastive divergence learning. Theoretic analysis reveals that X-MEN provably converges to the *global optimum* of the likelihood

function in linear number of SGD iterations. In addition, X-MEN can handle rewards parameterized either in a linear form or in the representation of a neural network. During testing, the policy learned by X-MEN can also be adapted to satisfying additional constraints without retraining.

In experiments, we compare the performance of X-MEN against MaxEnt IRL [Ziebart et al., 2008] and additional baselines such as Reletive Entropy IRL (RE-IRL) [Boularias et al., 2011] and recently proposed maximum likelihood constraint inference (MLCI) [Scobee and Sastry, 2019] on several grid world environments and in an imitation learning environment with human data to navigate around obstacles. All these environments require the agent to follow constraints. Our experiment shows after learning, the generated trajectories of X-MEN 100% satisfy constraints, while a majority of trajectories produced by competing approaches do not ($\geq 60\%$ violate constraints). Also X-MEN produces trajectories that closely imitate demonstrations. In summary, our contributions are as follows:

- We propose X-MEN, an algorithm that provably converges to the optimal reward function for MaxEnt IRL in linear number of training steps, even in the presence of multi-state combinatorial constraints.
- X-MEN is guaranteed to produce trajectories which satisfy combinatorial constraints, beyond the capability

of previous approaches.

- Experimental results reveal that X-MEN produces trajectories that closely resemble demonstration while satisfying constraints, outperforming a series of constrained IRL baselines.

# 2 INVERSE REINFORCEMENT LEARNING

Here we present a brief overview of IRL. $\mathcal{M} = \{\mathcal{S}, \mathcal{A}, T, R, \gamma\}$ is a Markov Decision Process (MDP), where $\mathcal{S}$ denotes the space of all states $s$, $\mathcal{A}$ denotes the set of possible actions $a$, $T$ denotes the transition probability function, $R$ denotes the reward function, and $\gamma \in [0, 1]$ is the discount factor. Given an MDP, an optimal policy $\pi^*$ is the one to maximize the expected cumulative reward. IRL considers the case where the reward function is unknown. Instead, a set of expert demonstrations $\mathcal{D} = \{\tau_1, \ldots, \tau_N\}$ is provided. Each demonstration consists of a series of state-action pairs $\tau_i = \{(s_{i1}, a_{i1}), \ldots, (s_{iL_i}, a_{iL_i})\}$, where $L_i$ denotes the length of the trajectory. The goal of IRL is to uncover the hidden reward $R$ from the demonstrations.

## 2.1 MAXIMUM ENTROPY IRL

A number of approaches have been proposed to tackle the IRL problem [Ng and Russell, 2000, Abbeel and Ng, 2004, Ratliff et al., 2006]. One crucial problem to address for IRL is to differentiate among multiple reward functions that lead to the same demonstrations. An influential formulation is Maximum Entropy IRL [Ziebart et al., 2008], which can also be viewed as a special case of Relative Entropy IRL (RE-IRL) [Boularias et al., 2011, Snoswell et al., 2020]. In this formulation, the probability that the demonstrator chooses a given trajectory is proportional to the exponent of the reward along the path. Denote $R_{\theta_1}(\tau) = \sum_{t=1}^{L} \gamma^t R_{\theta_1}(s_t, a_t)$ as the discounted cumulative reward parameterized by $\theta_1$. The probability of choosing trajectory $\tau$ is proportional to:

$$P_{choice}(\tau|\theta_1) \propto e^{R_{\theta_1}(\tau)}. \tag{1}$$

Let $d_0$ as the probability distribution of the initial state. $D(\tau) = d_0(s_1) \prod_{t=1}^{L} T(s_{t+1}|s_t, a_t)$ is the probability of state action transitions which leads to the trajectory $\tau$. Following the standard setup for (inverse) reinforcement learning, we assume $D(\tau)$ is unknown and needs to be learned from the interactions with the IRL system. For this paper, we parameterize $D(\tau)$ in the form of $e^{d_{\theta_2}(\tau)}$, where $\theta_2$ is the parameter to be learned. Hence, the overall probability of observing trajectory $\tau$ from demonstrations is proportional to the product of the choice probability and the state transition probability:

$$P(\tau|\theta, T) \propto e^{R_{\theta_1}(\tau)} D(\tau) = e^{R_{\theta_1}(\tau) + d_{\theta_2}(\tau)} = e^{R_\theta(\tau)}.$$

where $\theta = [\theta_1, \theta_2]$ is overall parameters to learn. We use $R_\theta(\tau)$ to represent $R_{\theta_1}(\tau) + d_{\theta_2}(\tau)$ with a slight overload of notations.

## 2.2 IRL WITH MULTI-STATE COMBINATORIAL CONSTRAINTS

Despite the success of many IRL models, many real world tasks require additional constraints to be satisfied when learning from demonstrations. In this work, we restrict ourselves to dealing with hard combinatorial constraints, as shown in Figure 1. Note that this is not particularly restrictive since, for example, safety constraints and/or constraints imposed by physical laws are often hard. Different from previous work that only defines constraints as a set of forbidden state-action pairs, which we call single-state constraints, here we consider more general cases of combinatorial constraints that span multiple states. Denote $C(\tau) = \{c_i(\tau)\}$ as the set of constraints that each trajectory must satisfy, and $I_C(\tau)$ the indicator function of whether constraints $C(\tau)$ are satisfied. Formally,

$$I_C(\tau) = \begin{cases} 1, & \text{if } \tau \text{ satisfies the constraints set } C(\tau) \\ 0, & \text{otherwise} \end{cases}$$

We augment the MDP into the constrained MDP: $\mathcal{M}^C = \{\mathcal{S}, \mathcal{A}, T, R, C\}$. In this case, the probability of observing a trajectory $\tau$ now becomes:

$$P(\tau|\theta, T) = \frac{1}{Z_\theta} e^{R_\theta(\tau)} I_C(\tau), \tag{2}$$

Here $Z_\theta$ is a normalization constant to ensure $P(\tau|\theta, T)$ is a probability distribution. Given the set of expert demonstrations $\mathcal{D}$, we want to find the best reward function by maximizing the log likelihood function $L(\theta)$.

$$\text{argmax}_\theta L(\theta) = \text{argmax}_\theta \frac{1}{|\mathcal{D}|} \sum_{\tau \in \mathcal{D}} R_\theta(\tau) - \log Z_\theta.$$

Notice only the terms related to the optimization variable $\theta$ are included in the rightmost equation.

# 3 XOR MAXIMUM ENTROPY IRL

In this section we propose XOR-Maximum ENtropy Constrained Inverse Reinforcement Learning (X-MEN), to solve the inverse reinforcement learning problem with multi-state combinatorial constraints. We develop X-MEN based on maximum entropy inverse reinforcement learning [Ziebart et al., 2008, Boularias et al., 2011, Finn et al., 2016]. Specifically, the model assumes that the expert samples the demonstrated trajectories $\{\tau_i\}$ from the distribution $P(\tau|\theta, T)$ in Equation 2, where $R_\theta(s_t, a_t) = \theta^T f(s_t, a_t)$ is represented by a linear combination of feature vector $f(s_t, a_t)$. $f(s_t, a_t)$

can be hand-crafted or generated by a deep neural network. Forward-backward dynamic programming can hardly solve this problem even if the state-transition function is given, due to the presence of the hard combinatorial constraints $I_C(\tau)$. Our X-MEN has the ability to solve this problem by leveraging XOR sampling to estimate $P(\tau|\theta, T)$. After learning, X-MEN will only take actions that lead to trajectories satisfying constraints.

We use Stochastic Gradient Descent (SGD) to optimize the objective, where in each iteration we compute the gradient of the log likelihood as follows:

$$\nabla_\theta L(\theta) = \frac{1}{|\mathcal{D}|} \sum_{\tau \in \mathcal{D}} \nabla_\theta R_\theta(\tau) - \nabla_\theta \log Z_\theta$$

$$= \frac{1}{|\mathcal{D}|} \sum_{\tau \in \mathcal{D}} \nabla_\theta R_\theta(\tau) - \sum_\tau P(\tau|\theta, T) \nabla_\theta R_\theta(\tau). \quad (3)$$

The first term in Equation 3 represents the expectation of $\nabla_\theta R_\theta(\tau)$ over all the trajectories in the training dataset, i.e., $\mathbb{E}_D[\nabla_\theta R_\theta(\tau)]$. The second term is the expectation of $\nabla_\theta R_\theta(\tau)$ over trajectories drawn from $P(\tau|\theta, T)$, i.e., $\mathbb{E}_P[\nabla_\theta R_\theta(\tau)]$. To approximate $\nabla_\theta L(\theta)$ in Equation 3, we sample $M_1$ trajectories from the dataset of demonstrations to form the set $\mathcal{D}_{M_1}$. Then we sample $M_2$ trajectories from $P(\tau|\theta, T)$, to form $\mathcal{D}_{M_2}^P$. We use $g_\theta$ in the following Theorem 1 to approximate $\nabla_\theta L(\theta)$:

**Theorem 1.** *Let the model distribution $P(\tau|\theta, T)$ defined in Equation 2 and $R_\theta(s_t, a_t) = \theta^T f(s_t, a_t)$. The gradient of the likelihood function defined in Equation 3. Let $g_\theta$ be*

$$g_\theta = \frac{1}{M_1} \sum_{\tau \in \mathcal{D}_{M_1}} f(\tau) - \frac{1}{M_2} \sum_{\tau \in \mathcal{D}_{M_2}^P} f(\tau), \quad (4)$$

*where $\mathcal{D}_{M_1}$, $\mathcal{D}_{M_2}^P$ are defined above. We must have $g_\theta$ is an unbiased estimation of $\nabla_\theta L(\theta)$, ie., $\mathbb{E}[g_\theta] = \nabla_\theta L(\theta)$.*

XOR-Sampling is used to obtain samples from $P(\tau|\theta, T)$ such that the probability of drawing a sample is sandwiched between a constant multiplicative bound of the true probability. XOR-Sampling is the result of a rich line of research [Ermon et al., 2013b, Gomes et al., 2006, 2007b], which translates the #-P complete sampling problem into queries to NP oracles with provable guarantees. The high level idea of XOR sampling is as follows. Suppose one would like to draw one ball uniformly at random from an urn, with access to an oracle that returns one ball from the urn once queried (implemented as an NP-oracle when sampling in a combinatorial space). Notice that the oracle will not return the balls uniformly at random; i.e., it may return the same ball every time. XOR-sampling removes the balls from the urn by introducing additional XOR constraints. One can prove that half of the balls are removed at random, each time when one XOR constraint is introduced. Hence, one keeps adding XOR constraints until there are only one ball remaining.

Then the last ball is returned. Since the balls are removed at random, the last left must be a random one drawn from the original set of balls. In practice, XOR-sampling also works with weighted probability distributions. While giving strong probabilistic guarantees, XOR-sampling requires solving NP-complete problems during the sampling process. Hence it introduces additional computational overhead compared to conventional approaches, e.g., MCMC sampling, etc. Nevertheless, recent advancements in constraint solvers allow us to solve industrial-sized combinatorial problems within reasonable amount of time. While we notice the trade-off between the computational overhead and the sample quality, we find the benefit of using XOR-sampling overweighs its cost in solving IRL problems involving hard combinatorial constraints.

Our paper uses the probabilistic bound of XOR-sampling via Theorem 2. We refer the readers to Ermon et al. [2013b], Ding et al. [2021], Ding and Xue [2021] for the details on the discretization scheme and the choice of the parameters of XOR-sampling to obtain the bound in Theorem 2.

**Theorem 2.** *[Ermon et al., 2013b] Let $\delta > 1$, $0 < \gamma < 1$, $w : \{0, 1\}^n \to \mathbb{R}^+$ be an unnormalized probability density function. $Q(\tau|\theta) \propto w(\tau)$ is the normalized distribution and $C(\tau)$ is the set of hard combinatorial constraints. Then, with probability at least $1 - \gamma$, XOR-Sampling$(w, C(\tau), \delta, \gamma)$ succeeds and outputs a sample $\tau_0$ by querying $O(-n \log(1 - 1/\sqrt{\delta}) \log(-n/\gamma \log(1 - 1/\sqrt{\delta})))$ NP oracles. Upon success, each $\tau_0$ is produced with probability $Q'(\tau_0)$. Let $\phi : \{0, 1\}^n \to \mathbb{R}^+$ be one non-negative function, then the expectation of one sampled $\phi(\tau)$ satisfies,*

$$\frac{1}{\delta} \mathbb{E}_Q[\phi(\tau)] \leq \mathbb{E}_{Q'}[\phi(\tau)] \leq \delta \mathbb{E}_Q[\phi(\tau)]. \quad (5)$$

The detailed procedure of X-MEN is shown in Algorithm 1. Here we demonstrate the version of X-MEN, where the only parameter to optimize is $\theta$. A variant of this algorithm can be developed which back-propagate the gradient over the feature vector $f(s, a)$ as well, when $f(s, a)$ is represented as a neural network and is also updated during learning. Notice when $f(s, a)$ is represented as a neural network, the log likelihood function is no longer concave. Hence the formal guarantees stated in Theorem 3 do not apply. However, this does not prevent X-MEN from being a useful algorithm in practice.

X-MEN takes as inputs the feature vector $f(s, a)$, transition probability $D(\tau)$, constraint set $C(\tau)$, training data $\{\tau_i\}_{i=1}^N$, initial model parameter $\theta_0$, the learning rate $\eta$, the number of SGD iterations $K$, XOR-Sampling parameters $(\delta, \gamma)$, and batch sizes $M_1$, $M_2$, and outputs the averaged learned parameter $\overline{\theta_K}$. To approximate $\mathbb{E}_P[\nabla_\theta R_\theta(\tau)]$ at the $k$-th iteration, X-MEN draws $M_2$ samples $\tau_1', \ldots, \tau_{M_2}'$ from $P(\tau|\theta, T)$ using XOR-Sampling, where $M_2$ is a user-determined sample size. Because XOR-Sampling has a

**Algorithm 1:** XOR Maximum Entropy Constrained Inverse Reinforcement Learning (X-MEN)

**Input:** $\theta_0, f(s,a), K, \eta, \delta, \gamma, D(\tau), C(\tau), M_1, M_2, \mathcal{D}$.

1 **for** $k = 0$ *to* $K$ **do**
2    $j \leftarrow 1$    // $M_1$ and $M_2$ are batch size
3    **while** $j \leq M_2$ **do**
4      $\tau' \leftarrow$ XOR-Sampling$\left(e^{\theta_k^T f(\tau)}, C(\tau), \delta, \gamma\right)$ **if**
     $\tau' \neq Failure$ **then**
5       $\tau'_j \leftarrow \tau'; j \leftarrow j + 1$
6      **end**
7    **end**
8    Get samples   $\mathcal{D}_{M_1} = \{\tau_j\}_{j=1}^{M_1}$ from $\mathcal{D}$.
9    $g_k = \frac{1}{M_1} \sum_{\tau \in \mathcal{D}_{M_1}} f(\tau) - \frac{1}{M_2} \sum_{j=1}^{M_2} f(\tau'_j)$
10    Update the parameters   $\theta_{k+1} = \theta_k + \eta g_k$
11 **end**
12 **return** $\overline{\theta_K} = \frac{1}{K} \sum_{k=1}^{K} \theta_k$

---

failure rate, X-MEN repeatedly call XOR-Sampling until all $M_2$ samples are obtained successfully (line $3-8$). Then, X-MEN also draws $M_1$ samples from the training set $\{\tau_i\}_{i=1}^N$ uniformly at random to approximate $\mathbb{E}_{\mathcal{D}}[\nabla_\theta R(\theta)]$. Once all the samples are obtained, X-MEN uses $g_k = \frac{1}{M_1} \sum_{\tau \in \mathcal{D}_{M_1}} f(\tau) - \frac{1}{M_2} \sum_{j=1}^{M_2} f(\tau'_j)$ as an approximation for the gradient of the negative log likelihood. $\theta$ is updated following the rule $\theta_{k+1} = \theta_k + \eta g_k$ for $K$ steps, where $\eta$ is the learning rate. Finally, the average of $\theta_1, \ldots, \theta_K$, namely $\overline{\theta_K} = \frac{1}{K} \sum_{k=1}^{K} \theta_k$ is the output of the algorithm. We show in the next sections that X-MEN enjoys the property of convergence to the global optimum of the log likelihood objective in linear number of iterations, and illustrate how to incorporate XOR-Sampling into our framework for sample generation with strict constraint satisfaction.

### 3.1 LINEARLY CONVERGE TO THE GLOBAL OPTIMUM

Suppose the only parameter to learn is $\theta$, in other words, $f(x, a)$ are fixed, the reward function $R_\theta(\tau)$ is represented by a linear combination of hand-crafted features, we can see that the objective is concave with regard to $\theta$. Under this circumstance, X-MEN converges to the global optimum of the log likelihood function in addition to a few vanishing terms. Moreover, the speed of the convergence is linear with respect to the number of stochastic gradient descent steps. Denote $Var_{\mathcal{D}}(f(\tau)) = \mathbb{E}_{\mathcal{D}}[||f(\tau)||_2^2] - ||\mathbb{E}_{\mathcal{D}}[f(\tau)]||_2^2$ and $Var_P(f(\tau)) = \mathbb{E}_P[||f(\tau)||_2^2] - ||\mathbb{E}_P[f(\tau)]||_2^2$ as the total variations of $f(\tau)$ w.r.t. the data distribution $P_{\mathcal{D}}$ and model distribution $P(\tau|\theta, T)$. The precise mathematical form of the convergence theorem states:

**Theorem 3.** *(main)*   *Let* $P(\tau|\theta, T)$ *be defined in Equation 2,* $R_\theta(\tau) = \theta^T f(\tau)$. *Given trajectories*

$\mathcal{D} = \{\tau_i\}_{i=1}^N$ *and the objective function* $L(\theta)$, *denote* $OPT = \max_\theta L(\theta)$ *and* $\theta^* = argmax_\theta L(\theta)$. *Let* $Var_{\mathcal{D}}(f(\tau)) \leq \sigma_1^2$, $||\mathbb{E}_{\mathcal{D}}[f(\tau)]||_2^2 \leq E^2$, $||\theta_k - \theta^*||_2 \leq R$, $\max_\theta Var_P(f(\tau)) \leq \sigma_2^2$, $||\mathbb{E}_P[f(\tau)^+]||_2^2 \leq G^2$, *and* $||\mathbb{E}_P[f(\tau)^-]||_2^2 \leq G^2$. *Suppose* $1 \leq \delta \leq \sqrt{2}$ *is used in XOR-sampling, the learning rate* $\eta \leq \frac{2-\delta^2}{\sigma_2^2 \delta}$, *and* $\overline{\theta_K}$ *is the output of X-MEN. We have:*

$$\mathbb{E}[L(\overline{\theta_K})] - OPT \leq \frac{\delta||\theta_0 - \theta^*||_2^2}{2\eta K} + \frac{\eta \sigma_1^2}{\delta M_1} + \frac{\eta \sigma_2^2}{\delta M_2} +$$
$$2(\delta^2 - 1)(G + E)R + 2\eta(\delta^3 - \delta)(G + E)^2.$$

X-MEN is the first provable algorithm which converges to the global optimum of the likelihood function and several tail terms for constrained inverse reinforcement learning problems. Moreover, the rate of the convergence is linear in the number of SGD iterations $K$. Previous approaches for IRL problems with hard combinatorial constraints do not have such tight bounds. In the bound stated above, the first term is inversely proportional to the number of SGD iterations $K$. The second and third terms can be minimized by increasing $M_1$ and $M_2$, i.e., with more samples drawn. The last two terms can be reduced by decreasing $\delta$, i.e., using more precise version of XOR-sampling.

The main challenge to prove Theorem 3 lies in the fact that we cannot ensure the unbiasedness of the gradient estimator. Because the objective is concave with respect to $\theta$ and smooth, a gradient descent algorithm can be proven to be linearly convergent towards the optimal value if the expectation of the estimated gradient is unbiased, ie, $\mathbb{E}[g_k] = \nabla_\theta L(\theta_k)$. However, even though we apply XOR-sampling, which has a constant approximation bound in generating samples from the model distribution, we still cannot guarantee the unbiasedness of $g_k$. Instead, using the constant factor approximation of XOR-Sampling, which is formally stated in Theorem 2, the bound for $g_k$ is in the following form

$$\frac{1}{\delta}[\nabla L(\theta_k)]^+ \leq \mathbb{E}[g_k^+] \leq \delta[\nabla L(\theta_k)]^+, \quad (6)$$

$$\delta[\nabla L(\theta_k)]^- \leq \mathbb{E}[g_k^-] \leq \frac{1}{\delta}[\nabla L(\theta_k)]^-. \quad (7)$$

Here, $\delta > 1$ is a constant factor, $[f]^+$ means the positive part of $f$, ie, $[f]^+ = \max\{f, \mathbf{0}\}$, and $[f]^-$ means the negative part of $f$, ie, $[f]^- = \min\{f, \mathbf{0}\}$.

The proof of Theorem 3 relies mainly on the following Theorem 4 which bounds the errors of Stochastic Gradient Descent (SGD) algorithms which only have access to constant approximate gradient vectors. Theorem 4 was proved in Ding et al. [2021], to help bound the errors of learning an exponential family model. Theorem 4 requires function $f$ to be $L$-smooth. $f(\theta)$ is $L$-smooth if and only if $||f(\theta_1) - f(\theta_2)||_2 \leq L||\theta_1 - \theta_2||_2$. Notice that the conditions of Theorem 3 automatically guarantee the $L$-smoothness of the objective and we leave the proof in the appendix.

**Theorem 4.** *[Ding et al., 2021] Let $f : \mathbb{R}^d \to \mathbb{R}$ be a $L$-smooth convex function and $\theta^* = argmin_\theta f(\theta)$. In iteration $k$ of SGD, $g_k$ is the estimated gradient, i.e., $\theta_{k+1} = \theta_k - \eta g_k$. If $Var(g_k) \leq \sigma^2$, $||\mathbb{E}[g_k^+]||_2 \leq G$, $||\mathbb{E}[g_k^-]||_2 \leq G$, $||\theta_t - \theta^*||_2 \leq R$, and there exists $1 \leq c \leq \sqrt{2}$ s.t. $\frac{1}{c}[\nabla f(\theta_k)]^+ \leq \mathbb{E}[g_k^+] \leq c[\nabla f(\theta_k)]^+$ and $c[\nabla f(\theta_k)]^- \leq \mathbb{E}[g_k^-] \leq \frac{1}{c}[\nabla f(\theta_k)]^-$, then for any $K > 1$ and step size $\eta \leq \frac{2-c^2}{Lc}$, let $\overline{\theta_K} = \frac{1}{K}\sum_{k=1}^{K}\theta_k$, we have*

$$\mathbb{E}[f(\overline{\theta_K})] - f(\theta^*) \leq \frac{c||\theta_0 - \theta^*||_2^2}{2\eta K} + \frac{\eta\sigma^2}{c} +$$
$$2(c - \frac{1}{c})GR + 2\eta(c - \frac{1}{c})G^2. \quad (8)$$

The proof of Theorem 3 is to apply Theorem 4 on the objective $L(\theta)$ and noticing that $L(\theta)$ is $L$-smooth when the total variation $Var_P(f(\tau))$ is bounded [Ding et al., 2021]. Theorem 3 states that in expectation, the difference between the output of X-MEN and the true optimum $OPT$ is bounded by a term that is inversely proportional to the number of iterations $K$ and several tail terms. In addition, to quantify the computational complexity of X-MEN, we prove the following theorem in the supplementary materials detailing the number of queries to NP oracles needed for X-MEN.

**Theorem 5.** *Let $|\mathcal{S}|$ and $|\mathcal{A}|$ be the number of all possible states and all possible actions, respectively, then X-MEN in Algorithm 1 uses $O(-K|\mathcal{S}||\mathcal{A}|\log(1 - 1/\sqrt{\delta})\log(-|\mathcal{S}||\mathcal{A}|/\gamma\log(1 - 1/\sqrt{\delta})) + KM_2)$ queries to NP oracles.*

## 4 RELATED WORK

Max-Ent IRL models were first proposed in [Ziebart et al., 2008] to addresses the inherent ambiguity of possible reward functions and induced policies for an observed behavior, during the training of which a forward-backward dynamic programming algorithm were used to exactly compute the partition function and marginal probability [Snoswell et al., 2020], assuming the knowledge of the transition probability. Relative Entropy IRL [Boularias et al., 2011] extends this work by leveraging an importance sampling approach to estimate the partition function unbiasedly without knowing the dynamics. Guided Cost Learning [Finn et al., 2016] further learns a Max-Ent model with policy optimization. Later work accommodates arbitrary nonlinear reward functions such as neural networks [Finn et al., 2016, Kalweit et al., 2020, Wulfmeier et al., 2015], instead of a linear combination of features. Recently proposed Generative Adversarial Imitation Learning (GAIL) [Ho and Ermon, 2016] is an imitation learning method that does not require estimating likelihoods. However, while Markovian rewards do often provide a succinct and expressive way to specify the objectives of a task, they cannot capture all possible task specifications, especially additional constraints [Vazquez-Chanlatte

et al., 2017]. Recent work on constrained IRL only focuses on local constraints of states, actions and features [Chou et al., 2018, Subramani et al., 2018, McPherson et al., 2018], which can hardly represent all the real world scenarios as most constraints are trajectory long. Other methods focus on learning constraints from the demonstrations, such as maximum likelihood constraint inference [Scobee and Sastry, 2019, Kalweit et al., 2020, Anwar et al., 2020, McPherson et al., 2021]. Our approach differs from all the existing methods and addresses the open question of learning with hard combinatorial constraints. We adapt the Max-Ent framework to allow us to reason about all the trajectories that satisfy the constraints during the contrastive learning process. Here we only consider pre-defined constraints. One should notice that even with the full knowledge of transition probability, dynamic programming cannot work well under trajectory-long constraints since it has no knowledge of any hard combinatorial information. X-MEN was motivated by the recent proposed probabilistic inference via hashing and randomization technique for both sampling [Ermon et al., 2013b, Ivrii et al., 2015], counting [Gomes et al., 2007a, Ding et al., 2019], and marginal inference problems [Ermon et al., 2013a, Kuck et al., 2019, Chakraborty et al., 2014, 2015, Belle et al., 2015] with constant approximation guarantees. Latest work also show the success of XOR-Sampling [Ermon et al., 2013b] to boost stochastic optimization algorithms [Ding and Xue, 2021] and improve machine learning tasks on structure generation [Ding et al., 2021].

## 5 EXPERIMENTS

We conduct experiments similar to those in Scobee and Sastry [2019], where we first show the superior performance of X-MEN in a synthetic grid world set of benchmarks. We also demonstrate the performance of X-MEN in mimicking trajectories from human participants as they navigate around obstacles and follow certain constraints on the floor. For comparison, we compare with classic Max-Ent IRL [Ziebart et al., 2008], RE-IRL [Boularias et al., 2011] and recently proposed maximum likelihood constraint inference (MLCI) Scobee and Sastry [2019] which can mask out the "not to go" states in the transition distribution. We implement X-MEN using IBM ILOG CPLEX Optimizer 12.63 for queries to NP oracles and XOR-Sampling parameters are same as Ding et al. [2021]. Experiments are carried out on a cluster, where each node has 24 cores and 96GB memory.

### 5.1 GRID WORLD

We consider a 9×9 grid world. The state corresponds to the location of the agent on the grid. The agent has three actions for moving up, right, or diagonally to the upper right by one cell. The objective is to move from the starting state in the bottom-left corner $s_0$ to the goal state in the up-right corner

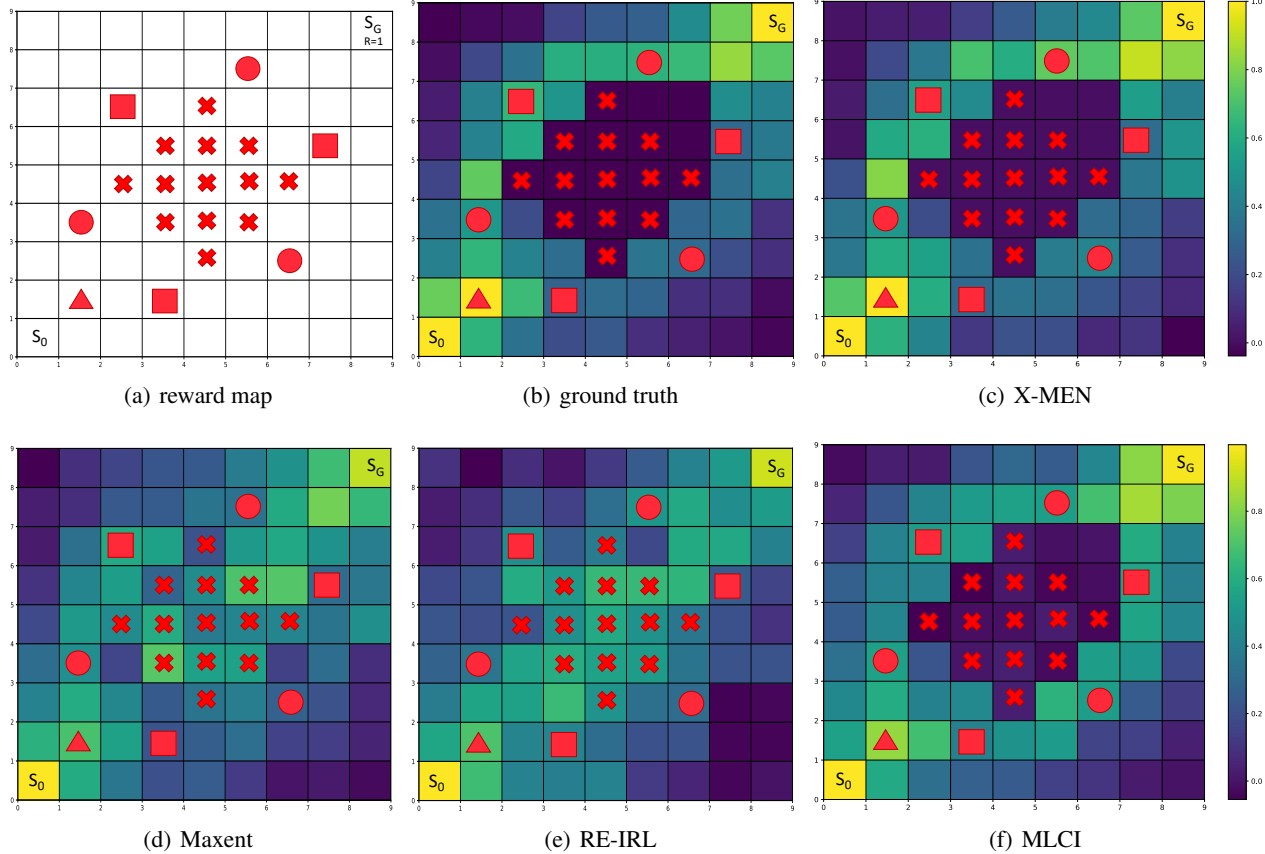

Figure 2: The superior performance of X-MEN against baselines in the grid world environment.**(a)** The ground truth reward map of the $9 \times 9$ gridworld. The reward of each state is 0, except for $S_G$ which is 1. Red symbols denotes constraints, where the red triangle denotes the state that must be passed through first among all the symbols, red crosses denote the states that can never be passed through, and the agent must pass through only one red square and one red circle. **(b)-(e)** The marginal probability of passing through each state of the ground truth demonstration and the distribution generated by different learning algorithms. We can see distribution of trajectories from X-MEN matches with the demonstration the most. Neither Maxent IRL nor RE-IRL can handle constraints. While MLCI knows "where not to go", it has difficulty in knowing "where must go" and we show in Figure 3 that it can not generate $100\%$ trajectories satisfying constraints.

$s_G$. Every state-action pair produces a distance feature, and the cumulative reward is inverse proportional to distance, which encourages short trajectories. There are additionally three more types of constraints, denoted as red symbols shown in Figure 2(a). The red triangle denotes the state that must be passed through first among all the symbols, red crosses denote the states that can never be passed through, and the agent must pass through only one red square and one red circle. The demonstration trajectories satisfies all the constraints and have an inductive bias: $70\%$ trajectories move along the upper paths and $30\%$ move along the lower path.

Due to the presence of hard constraints, recovering the reward map cannot be considered as the sole performance metric for a learning algorithm. In fact, an IRL agent with the groundtruth reward map may produce sub-optimal actions if he violates constraints. Therefore, we show in Figure

2(b)-2(f) the marginal distributions of passing each grid cell generated by aggregating 100 trajectories produced by different learning algorithms and the groundtruth demonstrations. We can see distribution of trajectories from X-MEN matches the demonstrations the most. Neither Maxent IRL nor RE-IRL can handle constraints. While MLCI knows "where not to go", it has difficulty in knowing "where must go" as the probability of the state marked as triangle is not 1 (we constrain that the agent must go through the triangle). Figure 3 further computes the percentage of valid trajectories generated by different algorithms varying the number of demonstration trajectories (3(a)) and training time (3(b)). X-MEN always generates $100\%$ valid trajectories while the competing methods satisfy no more than $50\%$. Moreover, we can see from the trend that even we keep increasing the number of demonstrations and the training time, the increase in baseline performance is minimal. Figure 3(c) compares

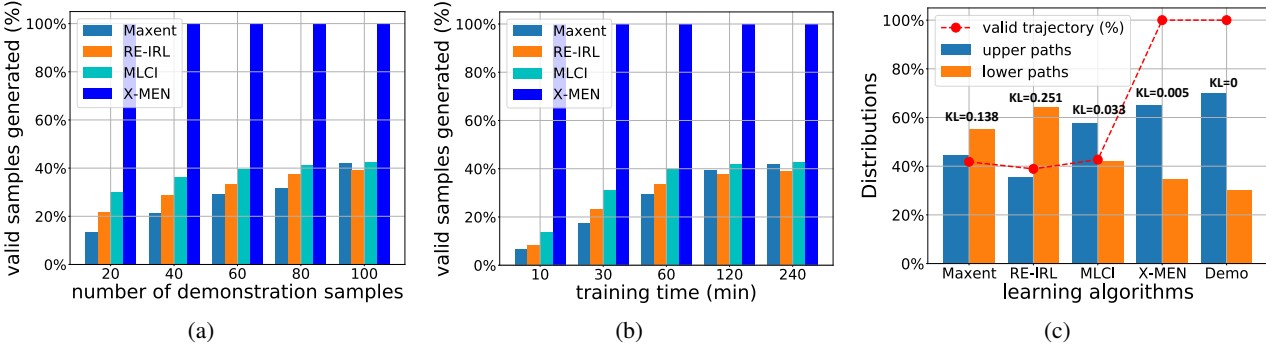

Figure 3: X-MEN outperforms competing approaches by producing 100% valid trajectories while capturing the inductive bias in demonstration on a $9 \times 9$ gridworld benchmark shown in Figure 2(a). (**Left**) The percentage of valid trajectories generated by different algorithms, varying the number of demonstration trajectories. (**Middle**) The percentage of valid trajectories generated by different algorithms varying training time. (**Right**) The dashed line shows the percentage of valid trajectories generated from different algorithms. The bars show the distributions of these valid trajectories grouped by different types of paths (upper paths or lower paths). X-MEN generates $100\%$ valid trajectories. The distribution of the trajectories has the minimal KL divergence $0.005$ towards that of the demonstrations.

the recovered distribution of the trajectories, where we can see X-MEN has the minimal KL divergence 0.005 towards the ground truth distribution of demonstration. The other baselines produce trajectories that significantly differ from the demonstrations (with larger KL-divergence).

## 5.2 HUMAN OBSTACLE AVOIDANCE

In our second example, we analyze trajectories from human beings as they navigate around obstacles on the floor and follow certain constraints. We map these continuous trajectories in a grid world where each cell represents a a 1ft-by-1ft area on the ground. The state corresponds to the location of the agent in the grid. The human agents are attempting to reach a fixed goal state $S_G$ from a given initial state $S_0$, as shown in Figure 4. The agent has only two actions for moving up or moving right. The shaded regions represent obstacles in the human's environment that cannot be passed through, and the red circle represent a "must pass" choke point that every person has to walk through. Additional hard constraints are that human cannot take the same action consecutively for more than 3 times.

Demonstrations were collected from 10 volunteers, who want to move from the start state to the goal state without violating any constraints. Empirical observations reveal that volunteers tend to follow the shortest paths given these constraints. We train both our model and the competing approaches using these demonstrations within the same training time of 4 hours and use 16 trajectory samples in each SGD iteration. Generated trajectories from X-MEN are shown in Figure 4, where we can see X-MEN is able to successfully avoid obstacles and pass the "must go" choke point. The 10 generated trajectories shown in the figure are

indeed the shortest paths from the start state to the goal (matching human demonstrations). Competing approaches do not generate trajectories that satisfy constraints, while the trajectories generated by X-MEN are $100\%$ valid. What worths noting is that X-MEN learns to go up first before passing through the gap between two obstacles, because otherwise the trajectory has to violate the constraint of taking the same action consecutively for more than 3 times.

## 6 CONCLUSION

We proposed X-MEN, a novel XOR maximum entropy framework for constrained Inverse Reinforcement Learning. We showed theoretically that X-MEN converges in linear speed towards the global optimum of the likelihood function for solving IRL problems. Empirically, we demonstrated the superior performance of X-MEN on two navigation tasks with additional hard combinatorial constraints. In all tasks, X-MEN generates 100% valid samples and the generated trajectories closely match the distribution of the training set. For future work, we would like to extend X-MEN to model-free reinforcement learning while preserving the theoretical guarantees. We also intend to test richer representations of the reward function in form of deep networks on real-world, large-scale constrained IRL tasks.

### Acknowledgements

This research is supported by NSF grant CCF-1918327.

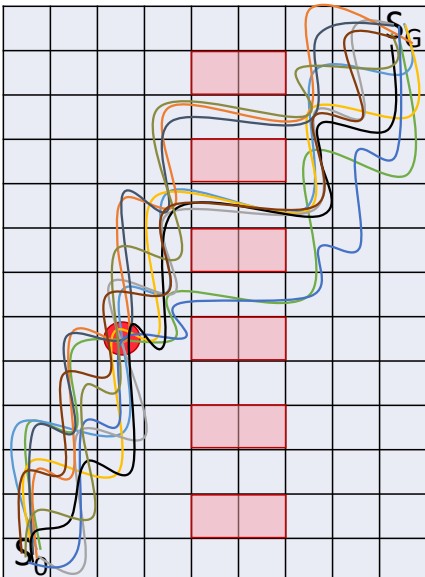

Figure 4: Overlaid trajectories generated by X-MEN after learning human preferences. The goal is to move from $S_0$ to $S_G$ and the action space contains only going up and right. The shaded regions represent obstacles in the human's environment, and the red circle represent a "must pass" point. Additional constraints are that human cannot take the same action consecutively for 3 times. We can see the generated trajectories from X-MEN satisfy all the constraints and follow the shortest possible paths, similar to what human demonstrators' actions.

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
