# OpenReview forum: "X-MEN: Guaranteed XOR-Maximum Entropy Constrained Inverse Reinforcement Learning"
_auai.org/UAI/2022/Conference — UAI 2022 Poster_

### Official Review · Reviewer_kwEY · 2022-04-11

**Q2(1) Originality/Novelty:** 3
**Q2(2) Significance/Impact:** 2
**Q2(3) Correctness/Technical Quality:** 3
**Q2(6) Clarity Of Writing:** 4
**Q6 Overall Score:** 6
**Q8 Confidence In Your Score:** 4

**Q1 Summary And Contributions:**

This paper focuses on IRL problems where certain multi-state combinatorial constraints are known beforehand. This paper proposes an algorithm based on MaxEnt IRL, which is guaranteed to produce trajectories satisfying combinatorial constraints, and proves that the proposed algorithm has a linear convergence to the global optimum. Reasonably designed experiments demonstrate the effectiveness of the algorithm.

**Q2 Assessment Of The Paper:**

More detailed information regarding each of these aspects is given below:

**Q2(4) Quality Of Experiments (Optional):**

3: Good: The experimental evaluation is adequate, and the results convincingly support the main claims.

**Q2(5) Reproducibility:**

2: Fair: Key resources (e.g., proofs, code, data) are unavailable but key details (e.g., proof sketches, experimental setup) are sufficiently well-described for an expert to confidently reproduce the main results.

**Q3 Main Strengths:**

1. This paper proposes an interesting problem of IRL.

2. The idea is relatively novel and the raised algorithm is effective as shown in the experiments.

3. The authors give a strong convergence analysis of the proposed algorithm, which provides theoretical support for their algorithm.


**Q4 Main Weakness:**

1. The authors only conduct experiments on a toy grid world task. Whether the algorithm can work efficiently when the state-action space is large or continuous, or the size of the constraints is large is still in doubt. As stated by the authors, it also remains a question that how to incorporate deep neural networks in their algorithms.

**Q5 Detailed Comments To The Authors:**

In equation 5, it should be Q = (1/Z) * (I_C(\tau)) * (e^{ \hat{\theta}^T f(\tau) }). Therefore, the related theoretical proof of Theorem 1 also should be modified.

**Q7 Justification For Your Score:**

I think this paper proposes a novel problem and gives an effective algorithm, although there are some minor errors, all of which are easy to correct. Overall, I think this paper is a good work and it contributes to the IRL community, so I give a week accept.

**Q9 Complying With Reviewing Instructions:**

1: Yes.

---

### Official Review · Reviewer_MXC8 · 2022-04-15

**Q2(1) Originality/Novelty:** 3
**Q2(2) Significance/Impact:** 3
**Q2(3) Correctness/Technical Quality:** 3
**Q2(6) Clarity Of Writing:** 3
**Q6 Overall Score:** 6
**Q8 Confidence In Your Score:** 3

**Q1 Summary And Contributions:**

This paper extends max-entropy IRL to account for hard multi-state combinatorial constraints.

**Q2 Assessment Of The Paper:**

More detailed information regarding each of these aspects is given below:

**Q2(4) Quality Of Experiments (Optional):**

2: Fair: The experimental evaluation is weak: important baselines are missing, or the results do not adequately support the main claims.

**Q2(5) Reproducibility:**

3: Good: Key resources (e.g., proofs, code, data) are available and key details (e.g., proofs, experimental setup) are sufficiently well-described for competent researchers to confidently reproduce the main results.

**Q3 Main Strengths:**

This work has extended max-entropy IRL to account for the important issue of hard multi-state combinatorial constraints.

This work has identified the use of XOR sampling that can theoretically guarantee the quality of reward parameters found by X-MEN.

The example in Figure 1 is very helpful towards understanding the motivation of this work.

**Q4 Main Weakness:**

There is a lack of discussion of the various possible practical examples (beyond the simple grid world and navigation tasks) to motivate the significance and usefulness of this work.

Unlike the extensive results given for the grid world, the details of the experimental setup, results, and analysis for the human obstacle avoidance experiment are too brief for a thorough understanding. Also, the experiments are performed on only toy settings.

Though the authors say on page 2 that their proposed X-MEN algorithm can handle rewards parameterized in the representation of a neural network and during testing, the policy learned by X-MEN can also be adapted to satisfying additional constraints without retraining, it is not clear from the main paper whether their experiments have done so. I assume that the theoretical analysis does not extend to nonlinear reward functions.

**Q5 Detailed Comments To The Authors:**

Are there any assumptions on the hard combinatorial constraints in the context of this work in order to preserve the probabilistic bound of XOR Sampling? If so, defining and describing them in more detail with examples (besides Figure 1c) would allow a reader to understand when it is suitable to use them and how to use them.

Can the authors discuss what the proposed algorithm would output in the case where the hard constraints cannot be satisfied?

Section 5.2 and Figure 4: I would have preferred that the authors use the remaining space in the main paper to include Figure 4 from the appendix or more details, results, and analysis. Furthermore, there is no color label in Figure 4 to indicate which paths belong to which tested algorithm.


Minor issues

Page 1: is located

Page 2: Reletive Entropy IRL

page 3: Let d_0 as

Page 6: to addresses

Page 7: inverse proportional

Page 8: Figure 1 in the appendix.


**Q7 Justification For Your Score:**

I like the contribution of this work in exploiting XOR sampling for theoretically guaranteeing the performance of the resulting algorithm.

The cons of this work are described in Q4.

POST REBUTTAL FEEDBACK

I like to thank the authors for their minor clarifications.

Since there were no substantial new findings/results to my concerns, I would maintain my current rating.

**Q9 Complying With Reviewing Instructions:**

1: Yes.

---

### Official Review · Reviewer_SuHY · 2022-04-17

**Q2(1) Originality/Novelty:** 2
**Q2(2) Significance/Impact:** 3
**Q2(3) Correctness/Technical Quality:** 3
**Q2(6) Clarity Of Writing:** 3
**Q6 Overall Score:** 7
**Q8 Confidence In Your Score:** 3

**Q1 Summary And Contributions:**

This paper introduces XOR-Maximum Entropy Constrained Inverse Reinforcement Learning (X-MEN), a method for inverse reinforcement learning.  X-MEN learns a reward function from trajectories taken from expert demonstrations in environments with hard constraints.  To do this, it utilizes XOR-sampling within the reward function optimization. Unlike previous work, it aims to support multi-state multi-action constraints over trajectories.


**Q2 Assessment Of The Paper:**

More detailed information regarding each of these aspects is given below:

**Q2(4) Quality Of Experiments (Optional):**

2: Fair: The experimental evaluation is weak: important baselines are missing, or the results do not adequately support the main claims.

**Q2(5) Reproducibility:**

3: Good: Key resources (e.g., proofs, code, data) are available and key details (e.g., proofs, experimental setup) are sufficiently well-described for competent researchers to confidently reproduce the main results.

**Q3 Main Strengths:**

The paper is well written, thorough, and tackles an interesting problem.

I believe it could serve as an important foundation for future work on IRL in more complex domains with more realistic constraints.

**Q4 Main Weakness:**

In places, I find the language used to describe what problem is solved by X-MEN is confused.  The IRL problem as stated in Section 2 is to infer the reward function $R$, which is what Algorithm 1 does.  But then in many places, the paper talks about learning an optimal policy.  These are different objectives, the gap between $R$ and an optimal policy (and any trajectories it generates) is, of course, an RL method, which as far as I can see is not discussed or defined.

I did not see any discussion on what limitations the XOR-Sampling imposes, both in terms of practical limitations and in what classes of trajectory predicates are permissible.  I understand many of the details are delegated to the respective papers, but since a central claim of the paper is increasing the expressiveness of constraints, there should be more information on what this means precisely.

**Q5 Detailed Comments To The Authors:**

- The main plot I would like to see is one showing divergence between the inferred $R$ and the ground truth $R$ as a function of the amount of data.  The results demonstrate that the method can handle the constraints, but (I think) what I actually care about is whether these constraints help me learn the reward function faster.

- The dataset consists of trajectories all of length $L$.  Would handling traces of differing lengths require a significant change in the method?

- The abstract claims "Empirical results in navigation demonstrate that X-MEN converges faster to the optimal policies compared to baseline approaches".  Again, I do not see anything in Algorithm 1 which outputs an optimal policy.  If I take this to mean it outputs the optimal reward function (according to your objective), what in the empirical results match this claim?

- How tied is the approach to the specific choice of the maximum entropy likelihood?

- A more common scenario to the one presented here is where the constraints are sometimes violated.  For example, one could imagine encoding the logical rules of a sports game.  It would be interesting to see if some adaption to this current mechanism could be made to handle this case, perhaps by having a hierarchical model that first chooses probabilistically between fully constrained vs unconstrained.  Ideally, even though the constraints are not always satisfied, they would serve as a strong inductive bias to help learn the true reward function with less data.

**Q7 Justification For Your Score:**

It makes a solid contribution.  I believe most of the weaknesses I outlined can be addressed.

**Q9 Complying With Reviewing Instructions:**

1: Yes.

---

### Decision · Program_Chairs · 2022-05-15

**Decision:**

Accept (Poster)

**Comment:**

Meta Review: The paper proposes a new inverse RL technique that takes into account hard multi-state combinatorial constraints.  This is a nice contribution that advances the state of the art since most inverse RL techniques do not take into account any constraint.  The authors are encouraged to follow the suggestions of the reviewers when preparing the final version of the paper.